# Fluorescence vs. Phosphorescence: Which Scenario Is Preferable in Au(I) Complexes with Benzothiadiazoles?

**DOI:** 10.3390/molecules27238162

**Published:** 2022-11-23

**Authors:** Radmir. M. Khisamov, Alexey A. Ryadun, Sergey N. Konchenko, Taisiya S. Sukhikh

**Affiliations:** Nikolaev Institute of Inorganic Chemistry, Siberian Branch of the Russian Academy of Sciences, 3 Lavrentiev Ave., 630090 Novosibirsk, Russia

**Keywords:** luminescence, Au(I) complexes, TD-DFT calculations, triplet state, single crystal X-ray diffraction, benzothiadiazole

## Abstract

The photoluminescence of Au(I) complexes is generally characterized by long radiative lifetimes owing to the large spin-orbital coupling constant of the Au(I) ion. Herein, we report three brightly emissive Au(I) coordination compounds, **1**, **2a,** and **2b**, that reveal unexpectedly short emission lifetimes of 10–20 ns. Polymorphs **2a** and **2b** exclusively exhibit fluorescence, which is quite rare for Au(I) compounds, while compound **1** reveals fluorescence as the major radiative pathway, and a minor contribution of a microsecond-scale component. The fluorescent behaviour for **1**–**2** is rationalized by means of quantum chemical (TD)-DFT calculations, which reveal the following: (1) S_0_–S_1_ and S_0_–T_1_ transitions mainly exhibit an intraligand nature. (2) The calculated spin-orbital coupling (SOC) between the states is small, which is a consequence of overall small metal contribution to the frontier orbitals. (3) The T_1_ state features much lower energy than the S_1_ state (by ca. 7000 cm^−1^), which hinders the SOC between the states. Thus, the S_1_ state decays in the form of fluorescence, rather than couples with T_1_. In the specific case of complex **1**, the potential energy surfaces for the S_1_ and T_2_ states intersect, while the vibrationally resolved S_1_–S_0_ and T_2_–S_0_ calculated radiative transitions show substantial overlap. Thus, the microsecond-scale component for complex **1** can stem from the coupling between the S_1_ and T_2_ states.

## 1. Introduction

Au(I) is a heavy d^10^ ion, whose coordination compounds have widespread applications, particularly in areas associated with light emission [1,2,3,4]. For these applications, understanding the origin of photophysical processes and the structure–property relationships is essential, as they play a decisive role in the characteristics of functional materials [5,6,7]. Au coordination compounds generally reveal a phosphorescent nature of luminescence (both conventional phosphorescence and delayed fluorescence) [8,9,10,11], which is tailored by the heavy atom effect. The latter promotes spin-orbit coupling (SOC) between singlet and triplet states, thus facilitating intersystem crossing (ISC) between them, which relaxes the spin ‘forbidden’ radiative transitions [12]. In general, the heavier the metal atom of a coordination compound, the stronger the SOC [13,14,15]. However, recent studies revealed that the presence of heavy atoms does not guarantee efficient ISC, while the nature of ligands may play a critical role in determining the radiation pathway [16,17,18]. To this end, the theoretical understanding of the relaxation processes is crucial, especially since conventional ISC between the S_1_ and T_1_ states is not necessarily the only pathway [19,20]. For instance, in a recently reported Au(III) complex, the S_1_ state couples with T_4_ and T_5_, resulting in phosphorescence as a major radiation pathway, while the relative Au(I) complex mainly demonstrates fluorescence due to the absence of triplet states energetically close to S_1_ [21]. The study of two Pt(II) complexes revealed the existence of two channels, S_1_–T_1_ and S_1_–T_2_; structural differences in the complexes result in a dominance of either one or the other channel, which induces their different luminescent behaviour [22]. Recently reported benzothiadiazole-based Au(I) compounds feature high quantum yields, reaching 97% in various organic matrices [23]. Notably, in the latter complexes, only a prompt fluorescence was observed.

Inspired by these encouraging examples, we study the luminescent properties of novel Au(I) complexes **1**, **2a**, and **2b** with two closely related ligands bearing benzothiadiazole (btd) moiety. In this context, amino-substituted btds open up a great scope for the design of photoactive materials: a slight variation of the composition and structure of btds results in a wide variation of energetic characteristics of electronic transitions, which is the key to fine-tuning the photophysical properties [24,25,26,27]. Herein, we chose two phosphorus-nitrogen ligands, comprising the NH-btd moiety as a chromophore, and PPh_2_ moiety as a readily accessible unit for the metal coordination. Synthesized compounds **1**–**2** exhibit prompt fluorescence, unusual for Au(I) complexes. We provide an interpretation of this phenomenon using quantum chemical DFT calculations.

## 2. Results and Discussion

### 2.1. Synthesis and Crystal Structure of the Compounds

The new compound *N*-(diphenylphosphino)-2,1,3-benzothiadiazole-4-amine (*PN*; Figure 1) was prepared by the reaction of 4-amino-2,1,3-benzothiadiazole and chlorodiphenylphosphine, in the presence of triethylamine as a base. The corresponding 1,3-phosphinoamine, *PCN*, was prepared by the one-pot condensation of diphenylphospine, benzaldehyde, and 4-amino-2,1,3-benzothaidiazole as reported recently [28]. Reactions of equimolar amounts of (THT)AuCl (THT–tetrahydrothiophene) with *PN* and *PCN* yielded complexes [Au(*PN*)Cl] (**1**) and [Au(*PCN*)Cl] (**2**), respectively. Changing the molar ratio of the phosphinoamines and Au to 1:2 produced the same complexes. For complex **2**, we obtained two polymorphs, **2a** and **2b**. The first one (**2a**) precipitated by slow concentration of a tetrahydrofuran (THF) solution under vacuum, while the second polymorph (**2b**) was formed from oversaturated oily THF solution. Both polymorphs can be obtained as a single phase, as evidenced by powder X-ray diffraction (XRD) analysis (Appendix A).

According to single-crystal XRD analysis, compounds **1**, **2a**, and **2b** exhibit a linear coordination environment of the metal by the Cl and P atoms (Figure 1). Except for these, Au(I) reveals neither intra- nor intermolecular contacts. No other specific intermolecular interactions were found. The geometry of molecule **2a** mainly differs from **2b** in the torsion angle {Au–P–C–N} (56.2° in **2a** versus 165.7° in **2b**). According to DFT calculations, the single molecules have similar Gibbs energy (the energy difference is less than 1 kJ·mol^−1^), while the calculated energy barrier (both in vacuum and in THF), corresponding to the transition from **2a** to **2b**, amounts to 30 kJ·mol^−1^ (Figure 2). This implies that the conformers **2a** and **2b** can easily transform to each other in the solution, which is corroborated by the presence of one set of signals in the corresponding ^31^P and ^1^H NMR spectra. Upon crystallization, either one or the other conformer is implemented, depending on a slight variation in the conditions; specifically, almost the entire amount of the dissolved complex can precipitate as pure phase **2a**.

### 2.2. Photophysical Properties and TD-DFT Calculations

The UV-Vis spectra were measured for **1**–**2** in the solid state (Figure 3, Table 1). The shape of the spectra is quite similar, with the exception that the long wavelength band for compound **1** is hypsochromically shifted as compared to **2a** and **2b**. According to TD-DFT calculations for the single molecules at their ground-state optimised geometry (Appendix A), the long wavelength band corresponds to the *S*_0_–*S*_1_ transition between HOMO and LUMO orbitals (Figure 4, Appendix A). The transition has a locally excited (LE) character with a notable contribution of an intraligand charge transfer (ILCT) from the periphery to the thiadiazole moiety, which is common for btd derivatives (Figure 4) [28,29,30,31]. In addition, compound **1** features a minor contribution of Au and Cl orbitals to the HOMO, which implies the presence of (M+X)LCT (metal- and halogen-to-ligand charge transfer). Upon photoexcitation, solids **1**–**2** exhibit a bright emission, with the absolute quantum yields of 30–34% (Table 1). The position of the bands does not depend on the excitation wavelength in the range of 300–400 nm (Appendix A). The luminescence spectra of compounds **2a** and **2b** reveal a hypsochromic shift of the emission band by ca. 450 and 100 cm^−1^, respectively, compared with free *PCN*. Compound **1** exhibits a more pronounced hypsochromic shift by ca. 2750 cm^−1^ compared with *PN*. The absorption and emission spectra of **2a** and **2b** in THF solution (Appendix A) are identical, suggesting that there is one emitting species in the solution.

Unexpectedly, the emission decay curves of solids **1**–**2** exhibit kinetics on the nanosecond scale, which yields the emission lifetimes of 10–20 ns (Appendix A). This is not typical for Au complexes, which usually reveal microsecond-scale kinetics owing to efficient SOC between the singlet and triplet states. The emission bands change negligibly upon cooling from room temperature to 77 K (Appendix A). Compounds **2a** and **2b** show exclusively prompt luminescence, while **1** features a weak afterglow signal after a time delay of 300 μs; the position of the corresponding band approximately coincides with that in the steady-state spectrum (Figure 5a). Thus, we can speculate that both prompt and delayed emission result from the same excited state. Since the microsecond-scale component (with an estimated lifetime of 100 μs) is not detected against the background of the nanosecond-scale kinetics, we assume it has a contribution of less than 1% of the overall emission.

One should note that the afterglow intensity decreases under constant irradiation of a sample of **1,** even in deaerated conditions. Within an hour, the signal drops by 10 times under a 450 W source of white light. Under the same conditions, the overall steady-state emission spectrum changes negligibly. This implies that the triplet states decompose due to processes in the solid, and are less photostable compared to the singlet excited states. The afterglow signal and its behaviour persist for samples from several parallel syntheses.

To gain insights into the peculiar fluorescent behaviour of compounds **1**–**2**, we performed TD-DFT calculations for the emission transitions from the singlet and triplet excited states (Table 1), as well as the SOC matrix elements between these states (Appendix A). For complexes **2a** and **2b**, the calculated SOC is weak (the coefficients are 10 cm^−1^ and less), which is in line with the observed absence of a phosphorescence. The calculated SOC for **1** is several times higher, which is still quite weak, but it enables the mixing of the singlet and triplet states. For the complexes, the brightest excitation transition in the long wavelength region is S_0_–S_1_, while the others (including S_2_ and S_3_) feature a much shorter wavelength and a 10 times lower magnitude of oscillator strength (Table 2). Therefore, we consider the S_1_–T_n_ channels as the most probable pathways. The calculated T_1_ energy level is widely separated from the S_1_ level (the difference between vertical S_0_–S_1_ and S_0_–T_1_ transitions of ca. 7000 cm^−1^). This results in a very slow ISC from the S_1_ sate and thus enables fluorescence from this state. A similar observation was observed for Pt complexes that also feature high ΔE(S_1_−T_1_) of 6000 cm^−1^ resulting in the fluorescence along with conventional phosphorescence [32].

For complex **1**, the T_2_ level locates energetically much closer to S_1_ than does T_1_. Vibrationally resolved calculated T_2_–S_0_ and S_1_–S_0_ emission spectra strongly overlap owing to the presence of a large number of vibrational sublevels of the S_1_ and T_2_ manifolds (Figure 5b). The comparison of energies at different geometry states (Figure 6) revealed that the potential energy surface (PES) for the S_0_ state has a local minimum at the T_2_ equilibrium geometry, while no imaginary frequencies were found for this geometry. As evidenced by the energy diagram, the PESs for the S_1_ and T_2_ states intersect: for the equilibrium T_2_-geometry, the energies for the S_1_ and T_2_ states show reverse order as compared to the equilibrium of the S_1_ geometry. Note also that the HOMO orbital for the relaxed T_2_ state differs from that for S_1_ state by a higher contribution of Au and Cl atoms (Appendix A). In addition, the SOC matrix elements between S_1_ and T_2_ at the T_2_ equilibrium geometry are almost twice as large as those at the S_0_ and S_1_ geometries (Appendix A). Therefore, we assume that the vibronic SOC [33] between the S_1_ and T_2_ states is a process responsible for ISC in complex **1**, and, consequently, for the minor microsecond-scale emission in the visible region. This process is much less favourable with respect to the S_1_–S_0_ radiation relaxation, thus resulting in the fluorescence as the major pathway. A portion of absorbed energy could release as a phosphorescence from the T_1_ state, but our calculations predict its manifestation in the infrared region. Thus, it is not detectable under the measurement conditions.

Complexes **2a** and **2b** exhibit a larger ΔE(S_1_−T_2_) compared to complex **1**, and the vibrational sublevels for the S_1_ and T_2_ manifolds show almost null overlap. In addition, the potential energy surfaces for S_1_ and T_2_ do not intersect. This may be the reason why **2a** and **2b** do not show an afterglow signal. Figure 6 represents energy diagrams that summarize the different behaviours arising from the differences in the excited states of complexes **1** and **2a**. For **2b**, (Appendix A) the diagram is similar to that for **2a**.

## 3. Materials and Methods

### 3.1. General

All manipulations for the syntheses were performed using the Schlenk technique and a glovebox. Solvents were purified using the standard technique and stored under an argon atmosphere. The 1,3-aminophosphine was synthesized as described recently [28]. Elemental analyses were performed on various MICRO cube instruments (Langenselbold, Germany) for C, H, N, and S elements. IR-spectra were recorded on a Fourier IR spectrometer FT-801 (Simex, Novosibirsk, Russia) in KBr pellets. ^1^H NMR spectra (500.13 MHz) and ^31^P NMR spectra (202.45 MHz) were obtained with a Bruker DRX-500 spectrometer (Madison, Wisconsin, USA) in dry CDCl_3_ at room temperature (RT); the solvent peak was used as an internal reference.

The diffuse reflectance UV-Vis spectra of the solid samples (Appendix A) were obtained with a Shimadzu UV-3101 spectrophotometer (Kyoto, Japan) at RT. Samples for the measurements were prepared by a thorough grinding of a mixture of the compounds under study (about 0.01 mol fraction) with BaSO_4_, which was also used as a standard. Spectral dependences of the diffuse reflectance were converted into spectra of a Kubelka-Munk function. Emission and excitation spectra of solids were recorded at RT on a Fluorolog 3 spectrometer (Horiba Jobin Yvon, Edison, NJ, USA) equipped with a cooled PC177CE-010 photon detection module (Horiba Jobin Yvon, Edison, NJ, USA) with a PMT R2658 photomultiplier (Horiba Jobin Yvon, Edison, NJ, USA). Excitation and emission spectra were corrected for source intensity (lamp and grating) and emission spectral response (detector and grating) by standard correction curves. For the measurements, powdered samples were placed between two nonfluorescent quartz plates. The emission decay curves were recorded on the same instrument using the TCSPC (time-correlated single photon counting) technique. A Horiba Nanoled laser diode (350 nm) was used as an excitation source. The curves were fitted by either one or two exponential decays; the fit yielded the emission lifetime values. Absolute PL quantum yields (QY) for the solids were recorded using the Quanta-φ device of Fluorolog 3. For the photostability experiment, solid sample **1** was constantly irradiated under Ar conditions with a full-range spectrum Xe lamp ushio uxl450s-0 (450 W; Horiba Jobin Yvon, Edison, NJ, USA) for 1 h.

The UV-Vis spectra of the solutions in 1 cm quartz cuvettes in an argon atmosphere were obtained with Agilent Cary 60 spectrometer (Santa Clara, California, USA). The emission and excitation spectra of the same solutions were recorded on an Agilent Cary Eclipse spectrometer.

### 3.2. Theoretical Calculations

The ground state geometries were optimized in a vacuum, without any constraints, at the DFT level using the PBE0 functional [34], def2-TZVP(-f) basis set, def2-ECP effective core potential (60 core electrons) for the Au atoms and D3(BJ) dispersion correction. The equilibrium geometries of the S_1_ (for **2a**) and T_2_ states (for comparison between the geometries, see Appendix A) were optimized by TD(TDA)-PBE0-D3BJ/def2-TZVP(-f), and the T_1_ state was optimized by UPBE0-D3BJ/def2-TZVP(-f) in the gaseous state, without any constraints. The equilibrium geometry of the S_1_ state for **2b** was optimized by TD-BHandHLYP-D3BJ rather than TD-PBE0-D3BJ, because the optimization with the PBE0 functional yielded a twisted structure (Appendix A), with an underestimated energy of fluorescence. Optimization at the TD-BHandHLYP-D3BJ/def2-TZVP(-f) level of theory provided more reliable geometry and emission energy. The Frank–Condon and emission energies for all complexes were then calculated at the TD(A)-PBE0/def2-TZVP(-f) level. Regular TD-DFT was used for the singlet states, and the Tamm–Dancoff approximation of TD-DFT was used for the triplet states due to triplet instability problems in TD-DFT [35]. For the acceleration of calculations resolution of identity for the Coulomb part (RI) and the chain of spheres for the Fock exchange (COSX), approximations were used with the corresponding auxiliary basis [36,37]. Correspondence to the minima of optimized geometries were verified by analytical harmonic vibrational frequency calculations for the ground states and T_1_ (UKS), and by numerical harmonic vibrational frequency calculations for the excited states.

The scalar-relativistic ZORA method was used with the SARC-ZORA-TZVP basis set for Au atoms to take into account relativistic effects in the photophysical properties; the ZORA-def2-TZVP basis set was used for the other atoms with SARC/J and AutoAux auxiliary basis sets for acceleration via the RIJCOSX algorithm. The quasi-degenerate perturbation theory was used for the calculation of spin-orbital coupling regarding the ZORA-TDA-DFT results [38]. The SOC integrals were calculated by the RI-SOMF(1X) method [39]. The calculations of SOC between states do not include vibronic coupling and non-adiabatic couplings. All geometry optimizations and TD(TDA)-DFT calculations were conducted using hte Orca 4.2.1 program. Molecular orbitals were visualized using ChemCraft 1.8.

Vibronically resolved fluorescence (from S_1_ state) and phosphorescence (from T_1_ and T_2_ states) spectra were calculated by the path integral approach implemented in the ESD subprogram of Orca 5.0.3. Low-energy normal modes were cut due to their anharmonicity (TCutFreq value is 100 cm^−1^). The line shape was set by VOIGT option, which is the mean product of the Gaussian and Lorentzian curves (Linew parameter is 20 cm^−1^ and Inlinew parameter is 100 cm^−1^).

### 3.3. Syntheses

#### 3.3.1. Synthesis of *PN*

At 0 °C, a solution of Ph_2_PCl (0.594 mL, 3.31 mmol) in toluene (5 mL) was added dropwise to a solution of NH_2_-btd (0.500 g, 3.31 mmol) and Et_3_N (0.922 mL, 6.62 mmol) in toluene (5 mL). The resulting mixture was allowed to warm up to RT and stirred overnight. The precipitate of Et_3_N·HCl was removed by filtration and rinsed four times with 3 mL of toluene. The solutions were combined, and all volatiles were evaporated. The residue was purified by column chromatography (silica gel, ethyl acetate:hexane = 1:10), followed by recrystallization from diethyl ether. Yield 0.876 g (79%). Calc. for C_18_H_14_N_3_PS (335.36): C 64.5, H 4.2, N 12.5, S 9.6. Found C 64.4, H 4.4, N 12.1, S 9.8. ^31^P{H} NMR (C_6_D_6_, *δ*, ppm): 28.4 (s). ^1^H NMR (C_6_D_6_, *δ*, ppm): 7.45–7.42 (m, 4H), 7.19–7.03 (m, 5H, 1H from btd-H + solvent), 7.06–7.03 (m, 7H), 5.96 (d, 1H). IR-spectrum (KBr, cm^−1^): 3281 s, 3050 m, 3000 w, 1599 w, 1541 s, 1487 s, 1433 s, 1383 s, 1290 m, 1082 s, 1044 m, 1024 w, 905 s, 850 m, 816 m, 739 s, 696 s.

#### 3.3.2. Synthesis of [Au(*PN*)Cl] (**1**)

A mixture of *PN* (25 mg, 0.0745 mmol) and (THT)AuCl (24 mg, 0.0745 mmol) was dissolved in CH_3_CN and was stirred at RT overnight to give a clear solution. The solvent was evaporated, and the oil residue was extracted by toluene in vacuum at 60 °C to opbtain a yellow crystalline product. Yield: 25 mg (74%). Calc. for C_18_H_14_AuClN_3_PS (567.78 g/mol): C 38.1, H 2.5, N 7.4, S 5.6. Found C 37.8, H 2.4, N 7.0, S 6.1. ^31^P{H} NMR (CDCl_3_, δ, ppm): 57.46 (s). ^1^H NMR (CDCl_3_, δ, ppm): 7.88–7.83 (m, 4H), 7.67–7.63 (m, 2H), 7.60–7.57 (m, 5H), 7.45 (dd, 1H, J_1_ = 9 Hz, 7.4 Hz), 7.13 (d, 1H, J = 7.3 Hz), 6.01 (d, 1H, J = 4.7 Hz). IR (cm^−1^): 3372 (m), 3045 (w), 2919 (w), 2850 (w), 2016 (w), 1993 (w), 1962 (w), 1940 (w), 1917 (w), 1890 (w), 1816 (w), 1728 (w), 1696 (w), 1642 (w), 1607 (w), 1545 (s), 1489 (m), 1436 (m), 1379 (s), 1298 (m), 1270 (w), 1180 (w), 1160 (w), 1109 (s), 1084 (s), 1043 (w), 1001 (w), 909 (s), 872 (m), 828 (w), 797 (w), 763 (w), 743 (s), 717 (w), 707 (w), 690 (s), 655 (w).

#### 3.3.3. Synthesis of [Au(P*CN*)Cl] (**2a**)

A mixture of *PCN* (50 mg, 0.118 mmol) and (THT)AuCl (38 mg, 0.118 mmol) was dissolved in THF (5 mL). The yellowish solution was stirred at RT overnight and concentrated under vacuum. A portion of crystals suitable for single-crystal XRD analysis was formed after exposure of the solution at 2 °C for 1 day. The crystals were separated and washed with diethyl ether. The mother liquor was further concentrated to isolate an additional portion of the compound. The fine powder was separated and washed with diethyl ether. Total yield: 75 mg (97%). Calc. for AuC_25_H_20_N_3_PSCl (657.90 g/mol): C 45.6, H 3.1, N 6.4, S 4.9. Found C 46.1, H 3.2, N 6.2, S 5.1. ^31^P{H} NMR (CDCl_3_, δ, ppm): 44.4 (s). ^1^H NMR (CDCl_3_, δ, ppm): 7.76 (dd, 2H), 7.52 (t, 1H), 7.46–7.38 (m, 5H), 7.30 (t, 2H), 7.25 (d, 2H), 7.20–7.12 (m, 5H), 6.31 (p, 1H), 6.03 (t, 1H), 5.68 (dd, 1H). IR (cm^−1^): 3353 (m), 3056 (m), 2912 (m), 2853 (w), 1971 (w), 1901 (w), 1804 (w), 1606 (w), 1550 (s), 1494 (s), 1434 (s), 1408 (s), 1364 (w), 1298 (s), 1183 (m), 1124 (w), 1105 (s), 1079 (w), 1026 (w), 1000 (w), 916 (w), 900 (m), 853 (m), 830 (m), 796 (m), 777 (w), 735 (s), 694 (s), 632 (w)

#### 3.3.4. Synthesis of [Au(*PCN*)Cl] (**2b**)

The compound is obtained by fast concentration of a solution of **2** in THF to give an oily sample. Subsequent exposure of the sample in a small amount of THF yielded a portion of crystals **2b**. To obtain bulk **2b**, a portion of the obtained crystals was sown on an oversaturated solution of the compound in THF. Calcd for AuC_25_H_20_N_3_PSCl (657.90 g/mol): C 45.6, H 3.1, N 6.4, S 4.9; found C 45.9, H 3.1, N 6.3, S 4.8. IR (cm^−1^): 3396 (m), 3067 (w), 3023 (w), 2903 (w), 1606 (w), 1552 (s), 1500 (s), 1436 (w), 1413 (m), 1368 (w), 1308 (m), 1198 (w), 1130 (w), 1104 (m), 1077 (w), 1030 (w), 1000 (w), 901 (m), 855 (w), 832 (w), 797 (w), 776 (w), 740 (s), 695 (s), 557 (m).

### 3.4. X-ray Data

Single-crystal XRD data for compounds **1**–**2** (Appendix A) were collected at 150 K with a Bruker D8 Venture diffractometer (Madison, Wisconsin, USA) with a CMOS PHOTON III detector (Bruker, Madison, Wisconsin, USA) and IμS 3.0 microfocus source (MoK_α_ radiation (λ = 0.71073 Å), collimating Montel mirrors; Incoatec GmbH, Geesthacht, Germany). The crystal structures were solved using the SHELXT [40] and were refined using the SHELXL [41] programs with OLEX2 GUI [42]. Atomic displacement parameters for non-hydrogen atoms were refined anisotropically. Hydrogen atoms were placed geometrically, with the exception of those in the amino groups, which were localized from the residual electron density map and refined with the restraining of the N–H bond (0.88 Å). The structures of **1**–**2** were deposited to the Cambridge Crystallographic Data Centre (CCDC) as a supplementary publication, No. 2217671-2217673.

Powder XRD data for the compounds (Appendix A) were collected at RT with a Bruker D8 Advance diffractometer in Bragg–Brentano geometry with an energy discriminating Eyger XE T detector (CuK_α_ radiation).

## 4. Conclusions

In summary, we synthesized Au(I) compounds **1**–**2** comprising closely related *PN* and *PCN* ligands. The latter comprise NH-btd moiety as a chromophore, and PPh_2_ moiety as a readily accessible unit for the metal coordination. The main difference between the ligands is that one of them (*PN*) contains a P–N bond, while in the other (*PCN*), the P and N atoms are separated by a methylene bridge (Figure 1). Such differences alter the nature of the luminescence of the corresponding complexes: polymorphs **2a** and **2b** exclusively exhibit fluorescence, which is not typical of Au(I) compounds, while compound **1** reveals fluorescence as the major radiative pathway, and a minor contribution of the microsecond-scale component.

We interpreted the unusual fluorescent behaviour using quantum chemical TD(A)-DFT calculations. In the complexes, the frontier molecular orbitals, responsible for the S_0_–S_1_ and S_0_–T_1_ transitions, are located almost exclusively on the *PN* and *PCN* ligands, i.e., the transitions mainly have an intraligand nature. Owing to the overall small metal contribution to the orbitals, the SOC is small. In addition, the T_1_ state has much lower energy than does the S_1_ (by ca. 7000 cm^−1^) state, which further hinders the ISC from S_1_. It thus enables the latter state to decay in the form of fluorescence as a dominant radiative process. In the specific case of complex **1**, the emission origin is not only limited to the prompt S_1_ decay, but also occurs with a mixing of singlet and triplet states, resulting in the presence of a minor microsecond-scale component. This behaviour is probably governed by a small contribution of (M+X)LCT, in the case of **1**, which is absent in **2a** and **2b**. As a possible channel, the SOC between the S_1_ and T_2_ states can contribute to ISC, owing to rich vibronic structure of the corresponding singlet and triplet manifolds.

## Data Availability

Not applicable.

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
