# Peer review of "Fluorescence vs. Phosphorescence: Which Scenario Is Preferable in Au(I) Complexes with Benzothiadiazoles?"

_molecules, 2022, doi:10.3390/molecules27238162_

Round 1
Reviewer 1 Report
In this manuscript, three Au(I) coordination compounds have been synthesized and fluorescent characterized, revealing unexpectedly short emission life times compared with the former reported Au-based compounds with phosphorescent nature of luminescence. The TD-DFT calculation has been used in this work to reveal the mechanism. I would like to see this work publish in this journal. Before acceptance, a minor revision is still needed.
1. In the fluorescence spectra for compound 1 in SI, Fig. S10, there is a minor peak at 600 nm. While in Fig. 3b, this peak disappears. Is there any impurity containing in this compound? Please give an explanation and revise this problem.
2. For better reading, the legend for Fig. 1 should be added; I suggest using (a), (b)… instead of the names of the compounds to label the figures 1, 4 and 6; (a) and (b) in Fig. 5 should be labeled; A rearrangement for Fig.4 and Table 1 is needed.
3. The bond length and angles selected from the single crystal data should be summarized in one table and added in SI file.
4. The format of the reference should be uniformed to satisfy the requirement of this journal, for instance, the abbreviation of the journal name in Ref. 21; the format for the titles of the references.
Author Response
We thank the Reviewer for the helpful comments and apologize for some inconsistencies. We have corrected minor inaccuracies and have revised the manuscript according to the recommendations. The point-by-point responses are as follows:
- In the fluorescence spectra for compound 1in SI, Fig. S10, there is a minor peak at 600 nm. While in Fig. 3b, this peak disappears. Is there any impurity containing in this compound? Please give an explanation and revise this problem.
In Fig. S10, a minor artefact at 600 nm for 1 is the second order Rayleigh scattering from the excitation lamp (the first order at 300 nm). The artefact shifts when we vary the excitation wavelength, which verifies its nature. We have clarified this in the caption to Fig. S10.
- For better reading, the legend for Fig. 1 should be added; I suggest using (a), (b)… instead of the names of the compounds to label the figures 1, 4 and 6; (a) and (b) in Fig. 5 should be labeled; A rearrangement for Fig.4 and Table 1 is needed.
We have added the legend for Fig. 1 and included the corresponding numeration in the figures. Fig. 4 and Table 1 were rearranged, so that Fig. 4 is now arranged in one page.
- The bond length and angles selected from the single crystal data should be summarized in one table and added in SI file.
We have added the corresponding data in SI (Tables S9–10 in the new version of SI)
- The format of the reference should be uniformed to satisfy the requirement of this journal, for instance, the abbreviation of the journal name in Ref. 21; the format for the titles of the references.
We have uniformed the references
Reviewer 2 Report
This is an interesting paper on the luminescence properties of two gold(I) complexes, denoted as 1 and 2, with ligands ‘PN’ and ‘PNP’, both comprising a benzodithiazol group chromophore and PPh2 unit for coordination. The crystal structures of the two complexes were determined by XRD (2 forms two polymorphs, 2a and 2b), The luminescence properties were determined by thorough photophysical measurements and the results were interpreted in terms of theoretical calculations on their electronic states. Overall, this is a valuable contribution to the photophysics of gold complexes. The paper is concise and clearly written. I recommend it for publication in Molecules, following some minor changes.
- row 36 ‘the heavier the atom’ is more clear as ‘the heavier the metal atom of the coordination compound’
- row 46 ’or another’ correctly ’or the other’
- row 62 ‘New N-(diphenylphosphino)-…’ correctly ’the new ligand N-(diphenylphosphino)-…”’
- I have not found the excitation wavelength in section 2.2, in which the photophysical properties of the complexes are presented.
- row 204 ‘Laser diode was used as an excitation source.’ Please, specify the type, wavelength and pulse length of the laser diode
Author Response
We thank the Reviewer for the helpful comments and apologize for some inconsistencies. We have corrected minor inaccuracies and have revised the manuscript according to the recommendations. The point-by-point responses are as follows:
- row 36 ‘the heavier the atom’ is more clear as ‘the heavier the metal atom of the coordination compound’
We have revised the phrase accordingly
- row 46 ’or another’ correctly ’or the other’
We have revised the phrase accordingly
- row 62 ‘New N-(diphenylphosphino)-…’ correctly ’the new ligand N-(diphenylphosphino)-…”’
We have revised the phrase accordingly
- I have not found the excitation wavelength in section 2.2, in which the photophysical properties of the complexes are presented.
Position of the emission bands does not depend on the excitation wavelength in the range of 300–400 nm; we have included this information in the text and have added the excitation spectra (Fig. S6 in the new version of SI). We also have specified the excitation wavelength used in each particular case in the captions to Figs. 3b, 5, and S7–S11. The quantum yields were measured at the excitation by 350 nm, as reflected in the caption to Table 1.
- row 204 ‘Laser diode was used as an excitation source.’ Please, specify the type, wavelength and pulse length of the laser diode
We used the Horiba Nanoled laser diode (350 nm); this information has been included in the Experimental section.
Reviewer 3 Report
Khisamov et al. presented a detailed photophysics study of three kinds of Au(I) complexes. One interesting finding is that complex 1 shows microsecond-scale photoluminescence, which stems from the S1-T2 coupling. All three kinds of complexes are well characterized by SCXRD. The authors also made some theoretical calculations to understand the photophysics of those complexes. I am very interested in the afterglow phenomena of complex 1. But I think the current discussion missed some details to support the proposed S1-T2 coupling mechanism. The following are some suggestions,
1. Please provide more details about the mentioned potential energy surface (PES) between S0 and T2. Meanwhile, the PES of complexes 2a and 2b are also missed.
2. Please provide the relaxed structures in the triplet state, while the orbital distribution in the triplet state is also suggested to provide. I believe the conformation change in the triplet state would play an important role in the afterglow phenomena.
Author Response
We thank the Reviewer for the helpful comments and apologize for some inconsistencies. We have corrected minor inaccuracies and have revised the manuscript according to the recommendations. The point-by-point responses are as follows:
- Please provide more details about the mentioned potential energy surface (PES) between S0 and T2. Meanwhile, the PES of complexes 2a and 2b are also missed.
We did not calculated full PES for different states due to the high computation cost. Instead, energies for the points corresponding to equilibrium S0, S1, T1 and T2 geometry states were calculated. These data allowed us to conclude that PESs for S1 and T2 states intersect. To avoid misunderstanding, we have revised the discussion with the emphasis that the conclusion was drawn from the analysis of the energy diagram.
- Please provide the relaxed structures in the triplet state, while the orbital distribution in the triplet state is also suggested to provide. I believe the conformation change in the triplet state would play an important role in the afterglow phenomena.
We have included a comparison of the relaxed geometries in S0, S1 and T2 states as Fig. S15 in the new version of SI. Frontier orbitals for the relaxed geometries of complex 1 were presented in Fig. S11 (S12 in the new version), while the differences in the orbital distribution (a higher contribution of Au and Cl atoms for T2 compared to S0 and S1) were mentioned in the text.